# Bitcoin Return Volatility Forecasting: A Comparative Study between GARCH and RNN

**Ze Shen** [1,*], **Qing Wan** [2] **and David J. Leatham** [1]

1    Department of Agricultural Economics, Texas A&M University, College Station, TX 77843, USA;
     d-leatham@tamu.edu
2    Department of Computer Science & Engineering, Texas A&M University, College Station, TX 77843, USA;
     frankqingwan@gmail.com
*    Correspondence: sz910201@tamu.edu; Tel.: +1-979-209-9468

**Abstract:** One of the notable features of bitcoin is its extreme volatility. The modeling and forecasting of bitcoin volatility are crucial for bitcoin investors' decision-making analysis and risk management. However, most previous studies of bitcoin volatility were founded on econometric models. Research on bitcoin volatility forecasting using machine learning algorithms is still sparse. In this study, both conventional econometric models and a machine learning model are used to forecast the bitcoin's return volatility and Value at Risk. The objective of this study is to compare their out-of-sample performance in forecasting accuracy and risk management efficiency. The results demonstrate that the RNN outperforms GARCH and EWMA in average forecasting performance. However, it is less efficient in capturing the bitcoin market's extreme events. Moreover, the RNN shows poor performance in Value at Risk forecasting, indicating that it could not work well as the econometric models in explaining extreme volatility. This study proposes an alternative method of bitcoin volatility analysis and provides more motivation for economic researchers to apply machine learning methods to the less volatile financial market conditions. Meanwhile, it also shows that the machine learning approaches are not always more advanced than econometric models, contrary to common belief.

**Keywords:** bitcoin; GARCH; machine learning; recurrent neural network; volatility; risk management

## 1. Introduction

Since Satoshi Nakamoto proposed the first cryptocurrency in 2009, the cryptocurrency market has received much attention. Bitcoin is the most successful and popular globally, accounting for over 50% of the whole cryptocurrency market capitalization in April 2019. Bitcoin's enthusiasm is due to its innovational features of decentralization and anonymity. Some public companies have started to hold bitcoin as an asset, and some financial institutions consider bitcoin as part of their investment strategy by allocating it in their portfolios. Many industries are interested in the blockchain technology behind bitcoin and have even started to launch their own cryptocurrencies. The agricultural industry is an excellent example, applying blockchain in agricultural insurance, product transactions, supply chain, and smart agriculture (Xiong et al. 2020). Covantis, a company co-owned by a global agribusiness group, has launched a blockchain platform for global commodities trading. Some cryptocurrencies are connected to agricultural industry trading in the market, for instance, Carboncoin, Blocery, and Herbalist Token. Although the application of blockchain technology and the cryptocurrencies related to agribusiness are in the infancy stage, it is still necessary for agribusiness researchers to understand this market. Therefore, examining insight into the bitcoin's behavior is a good starting point for understanding the Agri-crypto market. Researchers' analysis of bitcoin has received growing interest. David Yermack (2015) studied bitcoin's features and functions, and concluded that bitcoin appears to be more like a speculative investment than a real currency due to its high volatility.

So far, academic researchers have an agreement that bitcoin serves as a financial asset rather than a real currency. Therefore, it is necessary to look into its economic properties. There are two themes to explore: its price and risk.

People are excited about the tales of wealth that bitcoin can make a millionaire overnight. However, high return always goes with high risk. If people look at the bitcoin price history from 2009 until now, its violent fluctuations will be discovered. Investors have to be aware of these vast fluctuations and consider them when making investment decisions. As a financial asset, bitcoin is famous for its extreme volatility. Many factors could affect its fluctuation, i.e., transaction volume and frequency (Wątorek et al. 2021). Focusing on asset return volatility is a key to portfolio constructing, asset pricing, risk measuring, and managing. In particular, the asset return's volatility is a simple but widely used risk measurement. Researchers notice the fat tail of bitcoin return, indicating the high probability of significant losses. For better risk control, a sufficient amount of capital covering the potential losses of the asset trading at a given confidence level in the given period is required. This required value is called Value at Risk (VaR). A more accurate Value at Risk implies a higher risk management efficiency. The asset return's volatility is a key factor in Value at Risk calculation. Therefore, the modeling and forecasting of bitcoin volatility are crucial for bitcoin investors' decision-making analysis and risk management.

Economic researchers have been making efforts to improve the bitcoin return's volatility forecasting accuracy, and econometric methods are usually applied. Earlier studies mainly explored bitcoin volatility by using GARCH family models. Bouoiyour and Selmi (2015, 2016) compared different GARCH type models on sub-period bitcoin volatility, and Katsiampa (2017) compared the GARCH family models over the whole period. Balcilar et al. (2017) found the bitcoin trading volume fails to predict bitcoin volatility by studying their causal relationship. Troster et al. (2019) considered the heavy tail character of bitcoin return and compared bitcoin's return and volatility forecasting performance of GARCH and GAS model. They found that the heavy tail models outperform normally distributed models, and the heavy tailed GAS model provides better Value at Risk forecasting. Therefore, in terms of research techniques, the econometrics models are usually and maturely applied in bitcoin volatility forecasting.

However, research on bitcoin volatility forecasting using machine learning algorithms is still sparse. S. Athey (2019) pointed out that machine learning would dramatically impact the field of economics shortly. Unlike the economic models, where the researcher picks a specific model based on economic principles and estimates the parameters, a machine learning algorithm is usually data-driven modeling focused on the selection process. Thus, a machine learning model is not fixed or predetermined but will be refined during a training process. Applying machine learning methods to solve economic issues can potentially make a difference in the economic and financial fields. Some researchers have noticed this literature gap and started to apply machine learning approaches in cryptocurrency trading. A survey by Fang et al. (2021) indicates that up to 2019, among the research on cryptocurrency that involves technical methods, 13.8% applied machine learning methods. However, most of these researches use machine learning methods to predict cryptocurrency prices instead of volatility. Therefore, this study contributes to the literature gap by applying a machine learning method to bitcoin volatility forecasting. In this study, both conventional econometric models and a machine learning model are used to forecast bitcoin return volatility, and their forecasting performance is evaluated. This study aims to compare their performance and discover if machine learning can improve econometrics time series forecasting. The successful development of machine learning techniques in time series forecasting encourages people to apply them in the financial market. Moreover, machine learning's success in stock market prediction leads us to believe that it may also work well for cryptocurrency price forecasting. Also, the empirical studies show that the machine learning method is more efficient than the ARIMA model in bitcoin price prediction. McNally et al. (2018) compared the forecasting performance of the recurrent neural network (RNN), long short term memory (LSTM) network, and ARIMA on bitcoin price

and reported that the machine learning models outperformed ARIMA. Alessandretti et al. (2018) examined the forecasting performance on cryptocurrency portfolios and reported that machine learning methods overwhelm the standard benchmark simple moving average. It makes sense for the machine learning method to be superior to the traditional economic model (simple moving average and ARIMA). The machine learning model is proposed in a more general scope that considers both linear and nonlinear features. It also preserves more temporal information of a time series during training.

As discussed above, machine learning methods are more advanced than traditional economic models in time series forecasting theoretically and empirically. However, this assertion needs to be made cautiously. First of all, economic models involve economic intuition, while machine learning mainly deals with data. In the economic world, economic intuition is the key to economic analysis. In contrast, machine learning captures information only from data. However, the information contained in the data is limited in analyzing economic issues. Secondly, the performance of machine learning depends on the amount of data. Its performance is dramatically improved as the data amount getting larger. However, in this study, the bitcoin market history is relatively short. Finally, machine learning is sensitive to fluctuations. Compared to other approaches, machine learning is more efficient in identifying time-series trends and patterns. However, this leads to the problem that a shock or abnormal perturbation will be treated more seriously. Nevertheless, in the real world, many factors affect the market reaction to a shock or an abnormal perturbation, and the fluctuation sensitivity might cause overreaction problems in the forecasting, especially for the volatility analysis.

This study compares the forecasting performance between traditional econometric models and machine learning methods in forecasting accuracy and risk management efficiency. Investors are interested in the bitcoin volatility forecasting accuracy performance because they need information on how volatile the market will be in the future. Meanwhile, since the bitcoin market investors face significant risks every day, they are also concerned about risk management. This study contributes to the bitcoin volatility analysis literature in four ways. First, this is the first study that uses the RNN approach with GRU cell for bitcoin volatility forecasting. Second, the GARCH and the RNN are the most popular methods in their field. However, no research work gives detailed descriptions of how they perform differently in the bitcoin market. This study straightforwardly compares the two methods and gives a clear preference for applying either one under different situations. Third, in addition to volatility forecasting accuracy, this study also examines the Value at Risk efficiency, providing more applicable guidance for investors to implement in practice. Four, this study is not the first one to investigate whether the machine learning method is more advanced in financial time series forecasting. However, it contributes to the existing literature by providing more evidence on the limitations of applying machine learning approaches to solve economic issues.

This article is structured as follows. First, the econometric models are presented. It starts with the naive model, an exponentially weighted moving average (EWMA) as a benchmark, and then moves to a more complex but conventionally applied model, generalized autoregressive conditional heteroscedasticity (GARCH) model, to forecast bitcoin return volatility. Then a machine learning model based on Recurrent Neural Network (RNN) is proposed. The next step is to evaluate the out-of-sample performance of the three models. The root mean squared error (RMSE) and mean absolute error (MAE) are used to evaluate their forecasting accuracy performances, and the Value at Risk (VaR) is used to compare their risk management efficiency. Because bitcoin return's true conditional volatility is unobservable, the bitcoin daily squared return and Garman-Klass volatility (Garman and Klass 1980) are used as proxies for the realized volatility.

## 2. Materials and Methods

In this section, the econometric methodology is discussed first, and then the recurrent neural network model, which is a machine learning methodology, will be presented. Engle,

proposed the autoregressive conditional heteroscedasticity model (ARCH) in 1982, which assumes that the volatility of asset returns is time-varying instead of a constant. Bollerslev (1986) generalized the ARCH model and developed a more commonly used GARCH model. In this study, the GARCH model is applied as the econometric method.

The bitcoin return time series is used rather than the raw bitcoin price data. The bitcoin daily return is defined as the difference in the daily bitcoin closing price's natural logarithm. Bitcoin daily opening, high, low, and closing prices are used to estimate the realized bitcoin volatility. All the data are available on the website: CoinMarketCap.com, accessed on 8 July 2021. The data ranges from 30 April 2013 to 21 May 2021, with 2944 observations. Figure 1 illustrates the bitcoin daily return and bitcoin daily squared return respectively, and Table 1 shows the descriptive statistics of the bitcoin daily return.

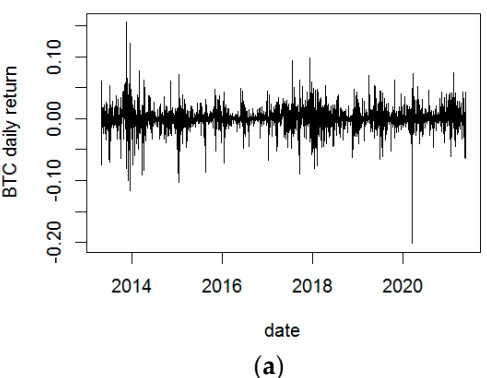
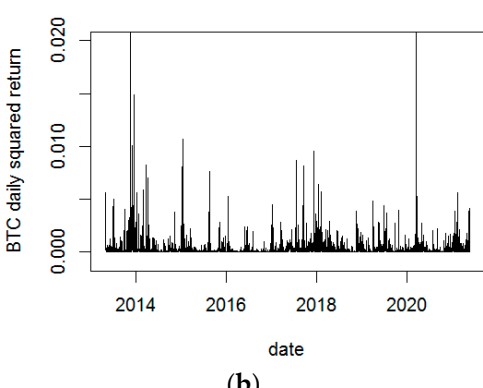

(**a**)               (**b**)

**Figure 1.** Bitcoin daily return and bitcoin daily squared return. (**a**) Bitcoin daily return; (**b**) bitcoin squared daily return.

**Table 1.** Summary Statistics of Bitcoin Daily Returns in the Sample Period.

| Description | BTC |
|:---:|:---:|
| Sample size | 2944 |
| Mean | 0.000819 |
| Variance | 0.000343 |
| Std. Dev. | 0.018524 |
| Skewness | −0.55305 |
| Kurtosis | 11.38964 |

Note: This table displays the summary statistics of bitcoin daily return from 30 April 2013 to 21 May 2021; BTC denotes bitcoin daily return.

Before going further to the econometric modeling, the stationary of the time series must be checked. The augmented Dickey-Fuller-test (ADF) and Phillips-Perron (PP) unit root test are used to check for the stationary of bitcoin daily return series, and Table 2 indicates that the financial time series is stationary.

**Table 2.** Unit Root Tests.

| | Without Trend | | With Trend | |
|:---:|:---:|:---:|:---:|:---:|
| | **ADF** | **PP** | **ADF** | **PP** |
| BTC daily return | −55.1 | −55.2 | −55.2 | −55.1 |
| Critical values (1%) | −3.43 | −3.43 | −3.96 | −3.96 |

Note: ADF and PP test statistics are much smaller than the 1% critical value, indicating the bitcoin daily return from 30 April 2013 to 21 May 2021 is stationary at 1% significance level.

### 2.1. Econometric Methodology

Figure 1 shows notable fluctuations in bitcoin daily return. It is also found that large changes follow large turbulence and small changes follow calm periods. This phenomenon

in time series asset return is known as "volatility clustering". The bitcoin daily squared return plot in Figure 1 provides more evidence that changes tend to be clustered together.

Table 3 shows the results of the Ljung-Box Q-test for the bitcoin daily squared return. Table 3 indicates that the bitcoin daily squared return is serially correlated, suggesting the existence of conditional heteroscedasticity in bitcoin price volatility. Thus, the econometric model needs to capture the feature of heteroscedasticity.

**Table 3.** Ljung-Box Q-Test for Bitcoin Daily Return.

| No. of Lags | Lag 10 | Lag 15 | Lag 20 |
|---|---|---|---|
| *p*-value | 0.000496 *** | 0.000693 *** | 0.000006 *** |

Note: Triple asterisk (***) denotes variable significant at 1% level.

The basic structure of the econometric model is as follows:

$$r_t = \mu_t + Z_t \tag{1}$$

$$\mu_t = E(r_t | F_{t-1}) \tag{2}$$

$$h_t^2 = Var(r_t | F_{t-1}) = E\left[(r_t - \mu_t)^2 | F_{t-1}\right] = E\left(Z_t^2 \Big| F_{t-1}\right) \tag{3}$$

where $r_t = \log \frac{p_t}{p_{t-1}}$, $\mu_t$ is the conditional mean, $h_t^2$ is the conditional variance, and $F_{t-1}$ denotes the past information.

### 2.1.1. Conditional Mean

ARMA $(p, q)$ process is applied to model the conditional mean:

$$\mu_t = \phi_0 + \sum_{i=1}^{p} \phi_i r_{t-i} + \sum_{j=1}^{q} \theta_j Z_{t-j} \tag{4}$$

with the autoregressive order $p$ and moving average order $q$.

After applying the ARMA $(p,q)$ process, the estimated parameters and the residuals are obtained. As discussed above, the bitcoin daily return exhibits volatility clustering, which indicates the conditional heteroscedasticity volatility. The ARCH effects of the residuals are tested. If there is an ARCH effect in the residuals, the conditional variance models will be specified in the next section.

### 2.1.2. Conditional Variance

Given the conditional mean model and using Equation (3), the residuals $Z_t = r_t - \mu_t$ are obtained. Then the conditional variance models can be built. Two different models are presented in the following section. It starts with EWMA model, then moves to the GARCH model to forecast bitcoin return volatility.

### 2.1.3. Exponentially Weighted Moving Average (EWMA)

The exponentially weighted moving average is one of the simplest models for volatility forecasting. It models the time-varying variance and captures past information and historical variance. Although the exponentially weighted moving average model incorporates neither conditional mean nor conditional variance in the sense of GARCH, it is presented here as a benchmark to evaluate the performance of the other models.

The exponentially weighted moving average model is presented as:

$$\sigma_{t+1}^2 = \lambda \sigma_t^2 + (1 - \lambda) r_t^2 \tag{5}$$

where $\lambda$ is set to be 0.94 in RiskMetrics model, and $r_t^2$ is the bitcoin daily squared return.

2.1.4. GARCH Model

Bollerslev developed the generalized autoregressive conditional heteroscedasticity model (GARCH) in 1986. Both the ARCH process and GARCH process model the variations of a financial assets' volatility, and the GARCH process allows the conditional variance to be an ARMA process. The GARCH process is as follows:

$$Z_t = h_t \varepsilon_t, \ \{\varepsilon_t\} \sim IID(0,1) \tag{6}$$

$$h_t^2 = \alpha_0 + \sum_{i=1}^{m} \alpha_i Z_{t-i}^2 + \sum_{j=1}^{n} \beta_j h_{t-1}^2 \tag{7}$$

where $\{Z_t\}$ is the residual series of the best-fitting ARMA $(p,q)$ model. Thus, the conditional variance of the residual series essentially acts like an ARMA process. It is expected that the standardized squared residuals obtained from the best fitted ARMA-GARCH model should not be autocorrelated, and there should not remain any ARCH effects. The ARCH LM test is used to check whether this is true or not in this study.

*2.2. Recurrent Neural Network (RNN)*

The sequencing model for predicting bitcoin return volatility is built on the concept of Recurrent Neural Networks (RNN). RNN deals well with sequence problems and has a remarkable architecture that considers the order of data. Each RNN has a type of memory unit concatenated into multi-stages and each of which will turn previous states and current input to activations and pass necessary information forward to the next stage. In this study, a GRU (Gated Recurrent Units) cell is employed to serve as the memory unit. The cost function is redesigned based on a tangent function. This model does not build any embedding or probability layer inside usual configurations in some engineering tasks. Besides, by considering some uncertainty of the volatility, the range is equally cut into 250 intervals to convert a real volatility value to a vector with a dimension of 250. This conversion serves as an encoder for an RNN cell's input. The whole architecture of the RNN model is listed in Figure 2. In general, the encoding process will turn a fixed length of sequential data into the same length of vectors for RNN, fed into multiple layers of perceptron (MLP). The MLP will decode states from RNN into sequential vectors and transfer them to a predictor for output.

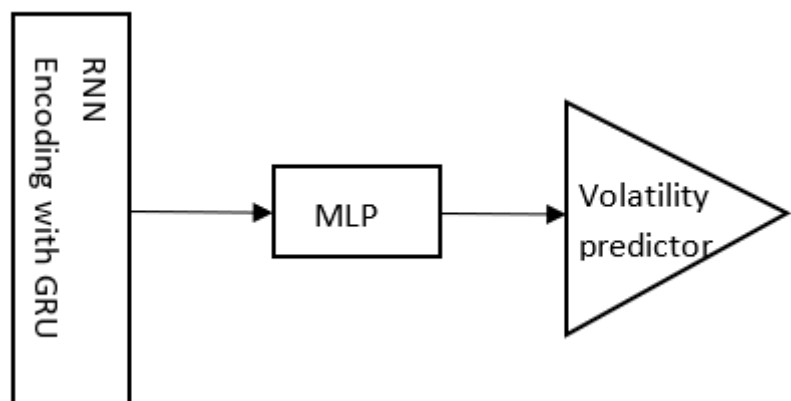

**Figure 2.** Architecture of recurrent neural network (RNN) model.

**3. Results**

In this section, the forecasting results of the EWMA model, GARCH model, and RNN model are presented. Then, their out-of-sample forecasting accuracy performance are evaluated and compared. Before the evaluation, appropriate proxies for the realized volatility have to be found.

*3.1. Forecasting*

The sample data is divided into two parts, the in-sample period from 30 April 2010, to 2 August 2020 (2652 observations) and the out-of-sample period from 3 August 2020, to 21 May 2021 (292 observations).

In the econometric GARCH model, the ARMA order is selected by AIC and BIC, and the best-fitted conditional mean model was found to be ARMA (2,2). Then, the residuals' ARCH effects are tested, and the result indicates there remains an ARCH effect in the residual series. Finally, the best-fitted ARMA-GARCH model is obtained. The ARMA-GARCH parameters are estimated in a rolling window. Table 4 presents the estimated parameters of the ARMA (2,2)-GARCH (1,2) model of the in-sample data.

**Table 4.** ARMA (2,2)-GARCH (1,2) estimated parameters.

| Parameters | Estimated Value | *t*-Value | *p*-Value |
|:---:|:---:|:---:|:---:|
| $\phi_0$ | 0.001 | 3.280 | 0.001 *** |
| $\phi_1$ | 1.424 | 128.130 | 0.000 *** |
| $\phi_2$ | $-0.434$ | $-39.606$ | 0.000 *** |
| $\theta_1$ | $-1.458$ | $-300{,}530$ | 0.000 *** |
| $\theta_2$ | 0.474 | 1452 | 0.000 *** |
| $\alpha_0$ | 0.000 | 1.851 | 0.064 * |
| $\alpha_1$ | 0.191 | 8.254 | 0.000 *** |
| $\beta_1$ | 0.413 | 2.965 | 0.003 ** |
| $\beta_2$ | 0.394 | 3.120 | 0.002 *** |

Note: Asterisk (*), double asterisk (**), and triple asterisk (***) denote variables significant at 10%, 5%, and 1% levels respectively.

The autocorrelation in the standardized residuals of the fitted ARMA-GARCH model is checked, and the result indicates that there is no remaining ARCH effect in the residuals.

For the RNN model, 30 days samples of the volatility were used to predict the next 1 day, 5 days, and 10 days with an out-of-sample method. For example, the first 30 days of volatility values were used to predict the 31st. The sequential data generated by this process is called tuple 1; then, the 2nd to 31st volatility values are used to predict the 32nd, and it is called tuple 2. The total data length was 2031. By rolling this process, 1994 tuples were generated. In the out-of-sample method, the first 1794 (90%) observations were appointed to training, and the remaining 200 observations were used as a test volume.

A more detailed implementation is illustrated in Figure 3. Two layers of RNN with GRU cells are built as a core. The first layer has 512 units, while the second shrink to 256 units. Sequential data were fed in cells on the bottom from left to right. The predicated data were collected on the top from left to right.

Both training and testing were taken on the GTX 1070 GPU. An SGD (Stochastic Gradient Descend) algorithm that shuffles the whole dataset is used in each iteration; the RMSProp gradient update algorithm was chosen as an optimizer; the learning rate and batch size were set to 0.0001 and 20, respectively. As stated before, the model 1000 epochs are trained on the 1794 tuples, and the 200 tuples are tested every five epochs.

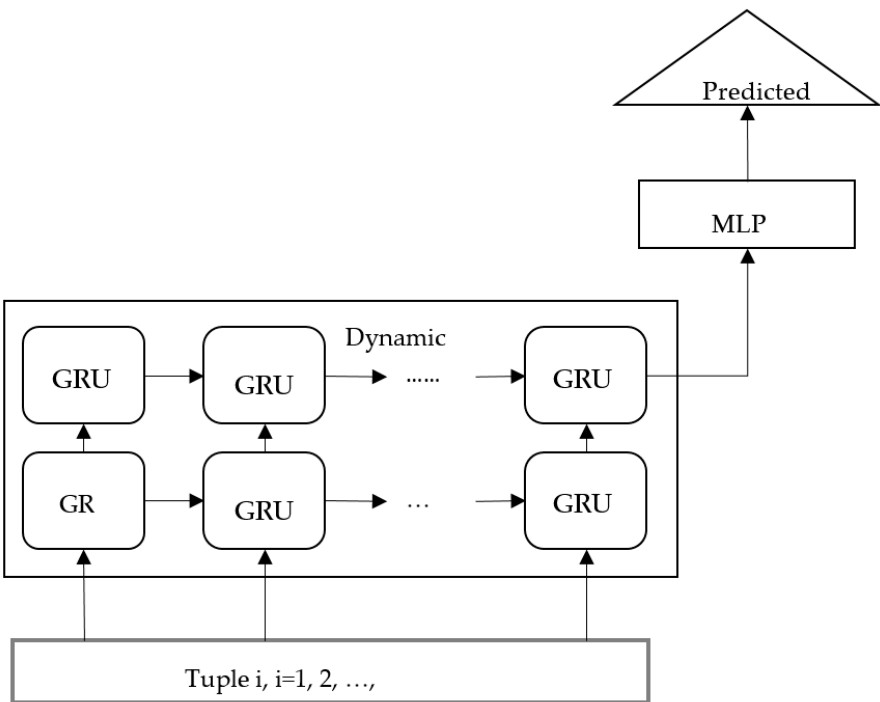

**Figure 3.** Detailed implementation of recurrent neural network (RNN) model.

### 3.2. Volatility Proxies

One difficulty evaluating the forecasting performance is that the true conditional volatility of the bitcoin return is unobservable. Thus, a proxy for the realized bitcoin return volatility has to be found. The most commonly used proxy for volatility is the bitcoin daily squared return. Thus, the first volatility proxy in this study is the daily squared return. However, it may lead to poor out-of-sample performance (Andersen and Bollerslev 1998). To get a more robust forecasting performance comparison result, a second volatility proxy is necessary. The cumulative squared intra-day returns are a more efficient proxy for volatility (Chou et al. 2010), but it requires high-frequency bitcoin prices in one day, which is not available in this case. Garman-Klass volatility (Garman and Klass 1980) is used as the second proxy for bitcoin return volatility. This proxy includes the information of daily high, low, opening, and closing prices. Garman and Klass (1980)'s estimator in practical is presented as:

$$\hat{\sigma}^2_{GK} = 0.5[\ln(BTC_{Ht}/BTC_{Lt})]^2 - [2\ln 2 - 1][\ln(BTC_{Ct}/BTC_{Ot})]^2 \qquad (8)$$

where $BTC_{Ht}$ and $BTC_{Lt}$ are the highest bitcoin price and lowest bitcoin price at the trading day, while $BTC_{Ct}$ and $BTC_{Ot}$ are the closing price and opening price, respectively.

### 3.3. Out-of-Sample Performance

To compare the three models' out-of-sample performance, the root mean squared error (RMSE) and mean absolute error (MAE) are used to evaluate and rank them. RMSE and MAE are the most commonly used metric for model evaluation. MAE is a good indicator of average model performance (Willmott and Matsuura 2005), while RMSE deals well with outliers by penalizing large errors more (Chai and Draxler 2014). Table 5 exhibits the WEMA (benchmark) model, GARCH model, and RNN model's out-of-sample performance. The out-of-sample performances at 1 day ahead, 5 days ahead, and 10 days ahead are presented in Table 5.

**Table 5.** Out-of-sample Performance.

| | RMSE | | MAE | |
|---|---|---|---|---|
| | **1st Proxy** | **2nd Proxy** | **1st Proxy** | **2nd Proxy** |
| *1 day ahead* | | | | |
| EWMA | $2.4797 \times 10^{-6}$ | $4.5816 \times 10^{-4}$ | $3.1640 \times 10^{-4}$ | $3.8195 \times 10^{-3}$ |
| GARCH | $\mathbf{2.4795 \times 10^{-6}}$ | $\mathbf{4.5814 \times 10^{-4}}$ | $3.3930 \times 10^{-4}$ | $\mathbf{3.7762 \times 10^{-3}}$ |
| RNN | $2.4985 \times 10^{-6}$ | $4.5818 \times 10^{-4}$ | $\mathbf{2.8091 \times 10^{-4}}$ | $4.0179 \times 10^{-3}$ |
| *5 days ahead* | | | | |
| EWMA | $2.4969 \times 10^{-6}$ | $4.6213 \times 10^{-4}$ | $3.1742 \times 10^{-4}$ | $3.8721 \times 10^{-3}$ |
| GARCH | $\mathbf{2.4760 \times 10^{-6}}$ | $\mathbf{4.5814 \times 10^{-4}}$ | $3.4605 \times 10^{-4}$ | $\mathbf{3.7607 \times 10^{-3}}$ |
| RNN | $2.5063 \times 10^{-6}$ | $4.5897 \times 10^{-4}$ | $\mathbf{2.8533 \times 10^{-4}}$ | $4.0535 \times 10^{-3}$ |
| *10 days ahead* | | | | |
| EWMA | $2.4779 \times 10^{-6}$ | $4.5817 \times 10^{-4}$ | $3.2214 \times 10^{-4}$ | $3.8277 \times 10^{-3}$ |
| GARCH | $\mathbf{2.4741 \times 10^{-6}}$ | $\mathbf{4.5813 \times 10^{-4}}$ | $3.5656 \times 10^{-4}$ | $\mathbf{3.7426 \times 10^{-3}}$ |
| RNN | $2.5065 \times 10^{-6}$ | $4.5897 \times 10^{-4}$ | $\mathbf{2.8426 \times 10^{-4}}$ | $4.0480 \times 10^{-3}$ |

Note: The lowest RMSE and MAE of each model is in bold.

Comparing the RMSE, the GARCH model performs best with the lowest RMSE, and the RNN model performs worst. When using the first proxy, the 1 (5, 10) day ahead RMSE of the RNN model are 0.76% (1.21%, 1.29%) larger than the GARCH model; when using the second proxy, the 1 (5, 10) day ahead RMSE of RNN model is 0.009% (0.18%, 0.18%) larger than the GARCH model. Comparing the MAE, the RNN model outperforms GARCH and EWMA with the lowest MAE in the first proxy but is outperformed in the second proxy. When using the first proxy, the 1 (5, 10) day ahead MAE of the GARCH model is 17.21% (17.55%, 20.28%) larger than the RNN model; when using the second proxy, the MAE of RNN is 6.02% (7.22%, 7.54%) larger than GARCH model. It can be seen from Table 5, when using the first proxy, RNN performs better in MAE but performs poorly in RMSE. One explanation for this is that since the RMSE punishes more on the outliers than MAE, implying the RNN model generates more outliers than the econometrics models. Figure 4 presents the standard deviation of the first proxy (daily squared return) and the standard deviation of one day ahead volatility forecasting of each model. Figure 5 shows the standard deviation of the second proxy (Garman-Klass volatility) and the standard deviation of one day ahead volatility forecasting of each model. EWMA1/5/10, GARCH1/5/10, and RNN1/5/10 denote the 1/5/10 day(s) ahead of the volatility forecasted by each model.

It can be seen from Figures 4 and 5 that the RNN model does better in capturing the volatility trends and clustering than the econometric models[1]; however, it underestimates the volatility. The second proxy is less volatile than the first one. Thus, the Garman-Klass volatility proxy does not perform as well as the bitcoin daily squared return proxy. The RNN model is not as efficient as we expected. It does better in corresponding to the volatility dynamics, but it underestimates the volatility and hurts the forecasting accuracy.

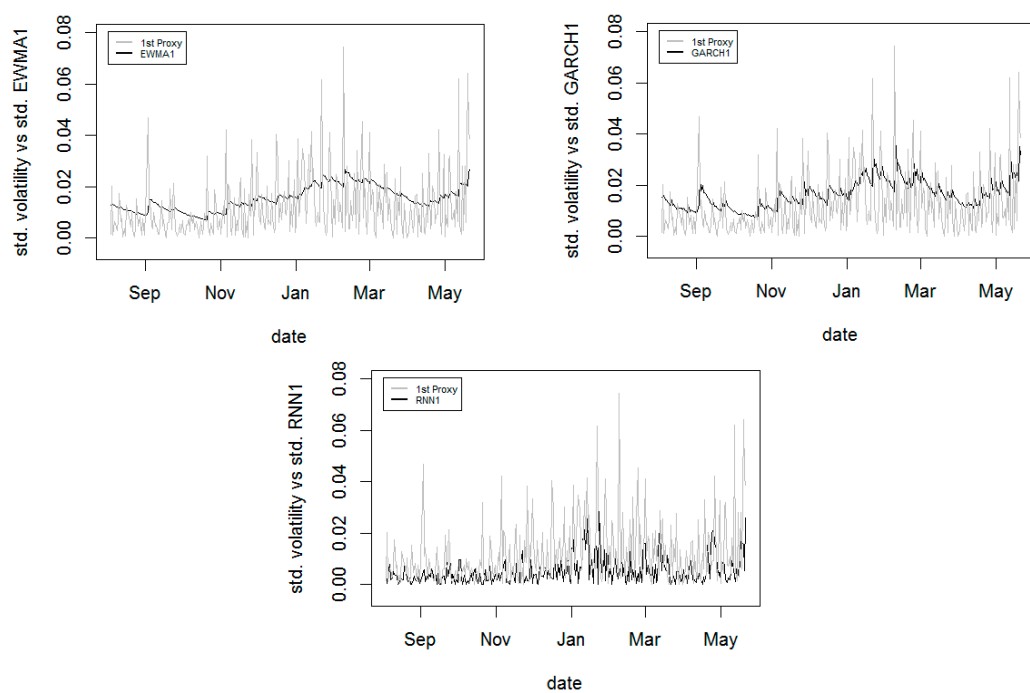

**Figure 4.** Out of sample standard deviation of realized volatility (1st proxy) vs standard deviation of one day ahead volatility forecasting of EWMA, GARCH, and RNN model. EWMA1, GARCH1, and RNN1 denote the one day ahead forecasting of each model.

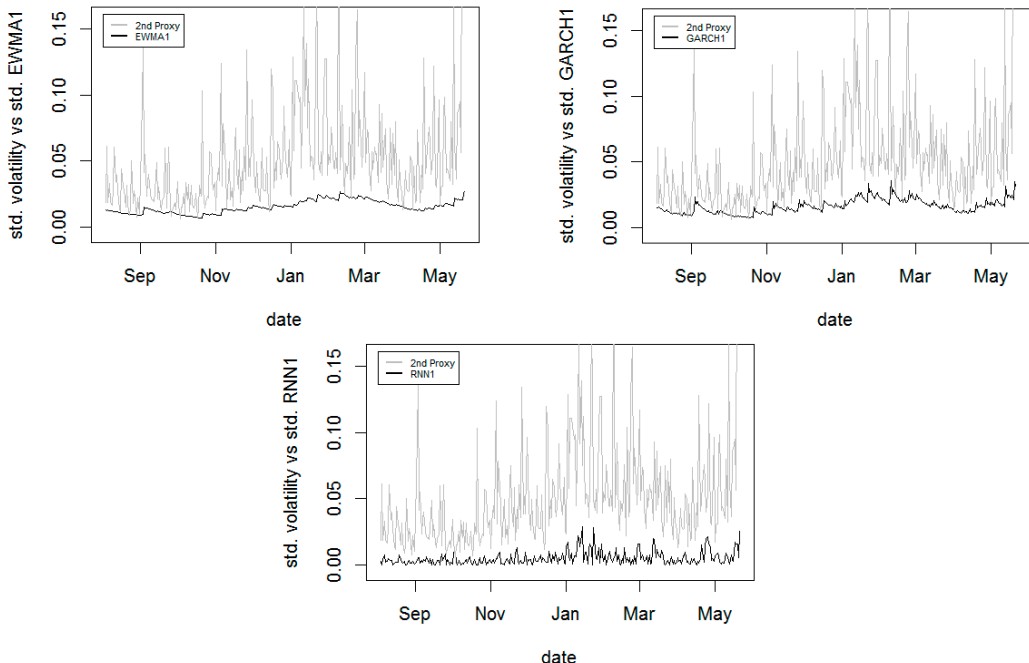

**Figure 5.** Out of sample standard deviation of realized volatility (2nd proxy) vs standard deviation of one day ahead volatility forecasting of EWMA, GARCH, and RNN model. EWMA1, GARCH1, and RNN1 denote the one day ahead forecasting of each model.

### 3.4. Value at Risk (VaR)

After analyzing the bitcoin volatility forecasting accuracy, it comes to the issue of risk management. Investors are also concerned about which method performs better in risk management. To provide more helpful information about the bitcoin market in terms of risk

management, the Value at Risk (VaR) is a suitable measurement. The bitcoin Value at Risk is defined as the maximum amount of money that the bitcoin investors could lose at a given confidence level over a defined period. J.P. Morgan proposed the Value at Risk concept in 1994. There are two categories of methodology for VaR calculation: parametric models and nonparametric models. The most used nonparametric VaR model is the Monto Carlo method, which is complicated, while the parametric VaR is simpler for investors to apply. The parametric models are mainly considered and discussed in this article. Mathematically, the forecasted VaR of bitcoin is defined as:

$$Pr(r_t \leq \text{VaR}_t(\alpha)|F_{t-1}) = \alpha \tag{9}$$

where $r_t$ is the bitcoin daily return, and $F_{t-1}$ is the past information. The $\alpha$ is the given confidence level, 1%, 2.5%, and 5%. Thus, the VaR estimation involves the assumption of bitcoin daily return distribution. The traditional VaR calculation assumes the portfolio return is normally distributed. However, it is empirically documented that financial asset return distributions always exhibit heavy tails. Table 1 lists the summary statistics for bitcoin daily returns, which shows there is excess kurtosis in the sample data. Therefore, the Student's t-distribution assumption is applied in this study instead of the normal one. The forecasting of daily bitcoin Value at Risk under Student's t-distribution is estimated as:

$$\text{VaR}_t(\alpha) = E(r_t|F_{t-1}) + t_\alpha^v \sigma_t \sqrt{\frac{v}{v-2}} \tag{10}$$

where $t_\alpha^v$ is the Student's t-distribution critical value at confident level $\alpha$ (1%, 2.5%, and 5%); $E(r_t|F_{t-1})$ is the conditional mean generated by ARMA (2,2); $\sigma_t$ is the standard deviation of volatility forecasted by the three models at time t. $v$ is the estimated degree of freedom. Following Heikkinen and Kanto (2002) and Andreev and Kanto (2004), the degree of freedom is allowed to be non-integer. Applying method of moments, the consistent estimator of the degree of freedom is estimated by:

$$\hat{v} = 4 + \frac{6}{\hat{k}}, \quad \forall v > 4 \tag{11}$$

where $\hat{k}$ is the sample excess kurtosis.

Since the EWMA model and RNN model do not involve conditional mean, then the $E(r_t|F_{t-1})$ of the two models are supposed to be zero. Therefore, the Value at Risk of EWMA and RNN models is estimated as:

$$\text{VaR}_t(\alpha) = t_\alpha^v \sigma_t \sqrt{\frac{v}{v-2}} \tag{12}$$

The sample excess kurtosis $\hat{k}$ is 8.39, then the estimated degree of freedom $\hat{v}$ is 4.72; the critical value $t_\alpha^v$ at confident level 1%, 2.5%, and 5% is $-3.45$, $-2.62$, $-2.04$, respectively.

Then the 1 day ahead, 5 days ahead and 10 days ahead VaR at 1%, 2.5%, and 5% confidence level is calculated by the conditional expected return and forecasted volatility, which are generated from the EWMA model, ARMA (2,2)-GARCH (1,2) model and RNN model, respectively. Figure 6 presents the realized returns and one day ahead VaR (1%)[2] forecasts for each of the three models.[3]

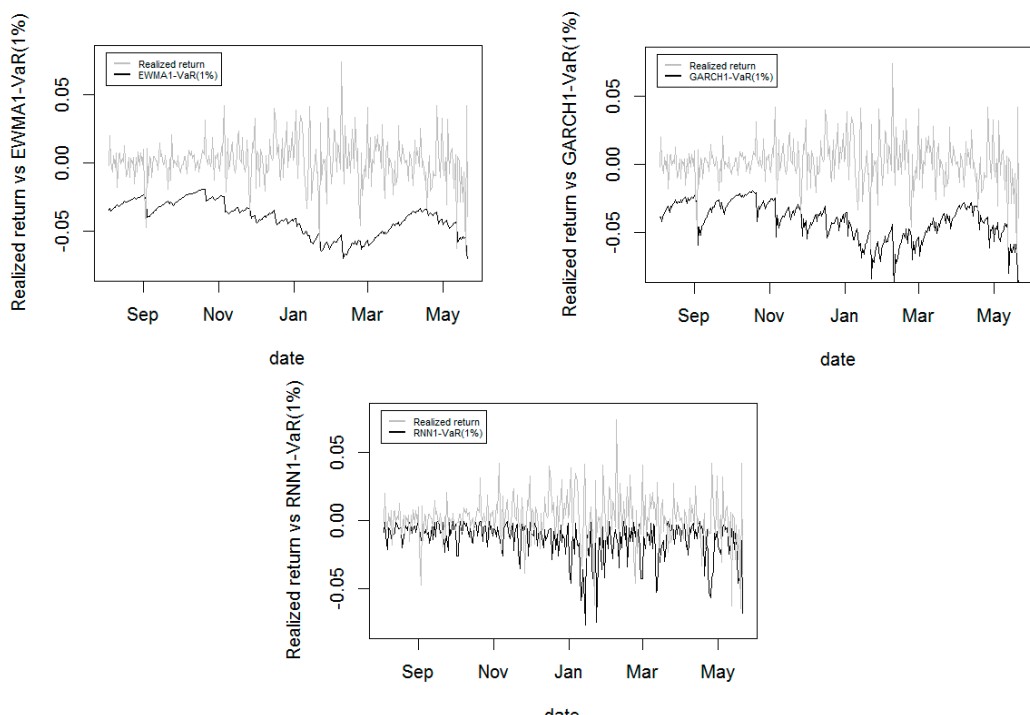

**Figure 6.** Out of sample realized return vs. one day ahead VaR (1%) of EWMA, GARCH, and RNN model. EWMA1, GARCH1, and RNN1 denote the one day ahead forecasting of each model.

Table 6 gives the out-of-sample coverage $\hat{\alpha}$ for each model to compare the Value at Risk forecasting performance. The $\hat{\alpha}$ is calculated by the number of realized losses that exceed the forecasted Value at Risk on that day divided by the number of totals out of sample observations:

$$\hat{\alpha} = \frac{No.\ (loss > VaR_t(\alpha))}{No.(out\ of\ sample\ obs.)} \tag{13}$$

**Table 6.** Out of Sample Coverage of Each Model.

|  | **EWMA** | **GARCH** | **RNN** |
|---|---|---|---|
| *1 day ahead* | | | |
| $\alpha = 1\%$ | **2.06%** | **2.06%** | 16.15% |
| $\alpha = 2.5\%$ | **2.75%** | **2.75%** | 18.9% |
| $\alpha = 5\%$ | 4.47% | **5.15%** | 23.37% |
| *5 days ahead* | | | |
| $\alpha = 1\%$ | **1.72%** | 2.06% | 21.31% |
| $\alpha = 2.5\%$ | 3.44% | **2.75%** | 25.43% |
| $\alpha = 5\%$ | 6.19% | **4.47%** | 26.80% |
| *10 days ahead* | | | |
| $\alpha = 1\%$ | **1.72%** | **1.72%** | 18.90% |
| $\alpha = 2.5\%$ | 3.78% | **2.41%** | 23.71% |
| $\alpha = 5\%$ | **5.16%** | 4.12% | 28.18% |

Note: The smallest $|\alpha - \hat{\alpha}|$ in each model is in bold.

The closer $\hat{\alpha}$ to $\alpha$, the more accurate VaR would be, making it easier for investors to manage the bitcoin market risk. Thus, the model with the smallest $|\alpha - \hat{\alpha}|$ provides the best risk coverage.

Table 6 indicates that the sample coverage $\hat{\alpha}$ of the GARCH model is closest to the given confidence level $\alpha$, while the sample coverage of the RNN model appears to be the

most volatile, with the largest distance between $\hat{\alpha}$ and $\alpha$. Thus, the RNN model performs poorly in Value at Risk forecasting, even worse than the benchmark EWMA model.

After getting the forecasted Value at Risk of each model, two approaches to Value at Risk backtesting are conducted: unconditional coverage test and conditional coverage test.

The unconditional coverage test was proposed by Kupiec in 1995. It tests whether the violation rate of the Value at Risk model is equal to the theoretical rate. Christoffersen introduced the conditional coverage test in 1998, which examines whether the VaR violation process is independent. Table 7 reports the VaR backtesting results of each model.

**Table 7.** VaR backtesting results of unconditional coverage test of Kupiec and conditional coverage test of Christoffersen *p*-value.

| | 1 Day Ahead | | 5 Days Ahead | | 10 Days Ahead | |
|---|---|---|---|---|---|---|
| | $LR_{uc}$ | $LR_{cc}$ | $LR_{uc}$ | $LR_{cc}$ | $LR_{uc}$ | $LR_{cc}$ |
| *0.99 Quantile* | | | | | | |
| EWMA | 0.11 | 0.25 | 0.26 | 0.49 | 0.26 | 0.49 |
| GARCH | 0.11 | 0.25 | 0.11 | 0.25 | 0.26 | 0.49 |
| RNN | 0.00 | 0.00 | 0.00 | 0.00 | 0.00 | 0.00 |
| *0.975 Quantile* | | | | | | |
| EWMA | 0.79 | 0.43 | 0.33 | 0.40 | 0.20 | 0.29 |
| GARCH | 0.79 | 0.43 | 0.79 | 0.43 | 0.92 | 0.84 |
| RNN | 0.00 | 0.00 | 0.00 | 0.00 | 0.00 | 0.00 |
| *0.95 Quantile* | | | | | | |
| EWMA | 0.67 | 0.80 | 0.37 | 0.67 | 0.90 | 0.94 |
| GARCH | 0.90 | 0.96 | 0.67 | 0.80 | 0.48 | 0.63 |
| RNN | 0.00 | 0.00 | 0.00 | 0.00 | 0.00 | 0.00 |

Note: $LR_{uc}$ denotes the *p* value of unconditional coverage Kupiec test and $LR_{cc}$ denotes the conditional coverage Christoffersen test. The *p*-value of the EWMA model and GARCH model show that they fail to reject the null hypothesis, indicating the VaR is correctly estimated. However, the RNN model fails in the two tests.

## 4. Discussion

Bitcoin is the most successful and popular cryptocurrency in the market, with around 130 billion daily trading volumes as of April 2019. Bitcoin has historically had more significant fluctuations in price than most other financial assets. Therefore, the analysis of bitcoin return volatility is crucial for investors' decision-making and risk management. Both economic models and the machine learning method are used to forecast the bitcoin return volatility.

The machine learning method in time series forecasting is expected to be superior to the traditional econometrics models. The earlier empirical studies in stock price forecasting and cryptocurrency prices forecasting provided evidence of this statement. By comparing the out-of-sample performance of each volatility forecasting model, the result indicates that the RNN model is more sensitive and corresponds more quickly to the volatility change than the traditional econometrics models. The RNN outperforms GARCH and EWMA in MAE evaluation criteria in forecasting accuracy but is overwhelmed in RMSE criteria. Since MAE does well in average model performance, while RMSE provides more information on outliers, the two opposite performances could be regarded as evidence that the RNN model is less efficient in capturing the bitcoin market extreme events.

In addition to the bitcoin volatility forecasting, the RNN model is outperformed by the GARCH and EWMA model in risk management efficiency in the framework of Value at Risk. The Value at Risk essentially focuses on the tail events of bitcoin return. Therefore, the RNN's poor performance in VaR provides another evidence of the robust results that it could not work well as econometric models in explaining extreme volatility. It underestimates the fluctuations in the more volatile price period. In other words, it underestimates

the risk. This result is consistent with the previous study that Seq2Seq RNNs improve the bitcoin price forecasting accuracy over the ARIMA model during less volatile periods but shows poor performance in extreme cases (Rebane et al. 2018).

This study proposed an alternative way of volatility analysis. It is widely believed and empirically proved by earlier studies in the financial market that the machine learning approach is more advanced in time series forecasting. However, this study shows something different. The RNN model is less efficient than traditional econometric models in bitcoin volatility forecasting and risk management. The econometric models are superior in analyzing extreme market conditions, while the machine learning approach is more suitable for less volatile market conditions.

Many investors have considered including bitcoin in their investment portfolio. This study provides implications for the investors on trading strategy and risk management. For the financial institutions required to hold sufficient risk capital to cover potential losses on the portfolio, the econometric models are recommended for a good Value at Risk estimation.

**Author Contributions:** Conceptualization, Z.S., Q.W. and D.J.L.; Formal analysis, Z.S. and Q.W.; Funding acquisition, D.J.L.; Investigation, Z.S.; Methodology, Z.S. and Q.W.; Project administration, D.J.L.; Resources, Z.S.; Software, Z.S. and Q.W.; Supervision, D.J.L.; Validation, Z.S. and Q.W.; Visualization, Z.S.; Writing—original draft, Z.S.; Writing—review & editing, Z.S., Q.W. and D.J.L. All authors have read and agreed to the published version of the manuscript.

**Funding:** This research received no external funding.

**Conflicts of Interest:** The authors declare no conflict of interest.

**Appendix A**

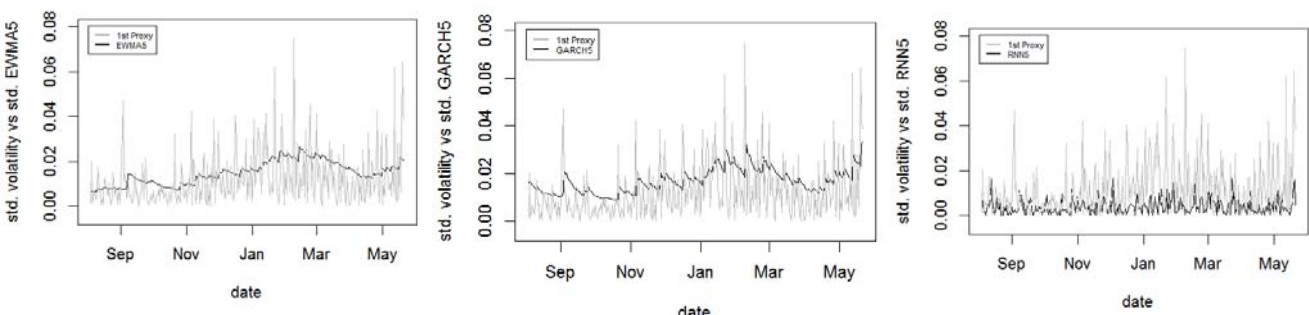

**Figure A1.** Out of sample standard deviation of realized volatility (1st proxy) vs. standard deviation of 5 days ahead volatility forecasting of EWMA, GARCH and RNN model.

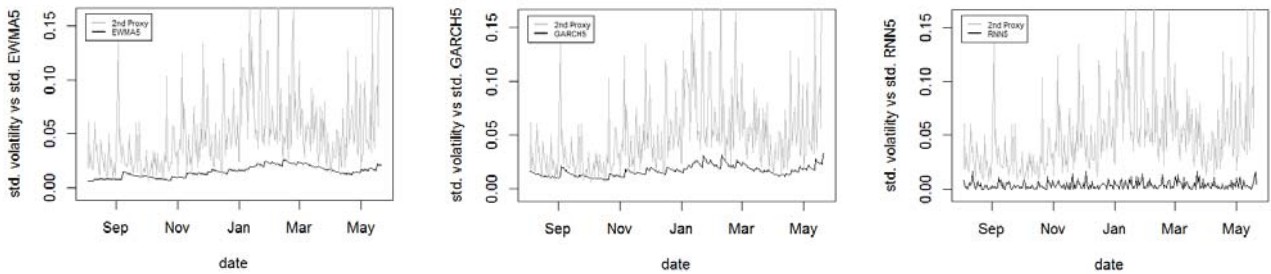

**Figure A2.** Out of sample standard deviation of realized volatility (2nd proxy) vs. standard deviation of 5 days ahead volatility forecasting of EWMA, GARCH and RNN model.

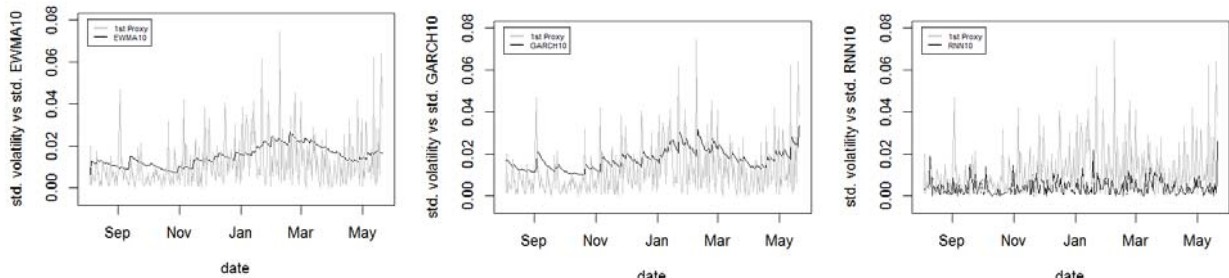

**Figure A3.** Out of sample standard deviation of realized volatility (1st proxy) vs. standard deviation of 10 days ahead volatility forecasting of EWMA, GARCH and RNN model.

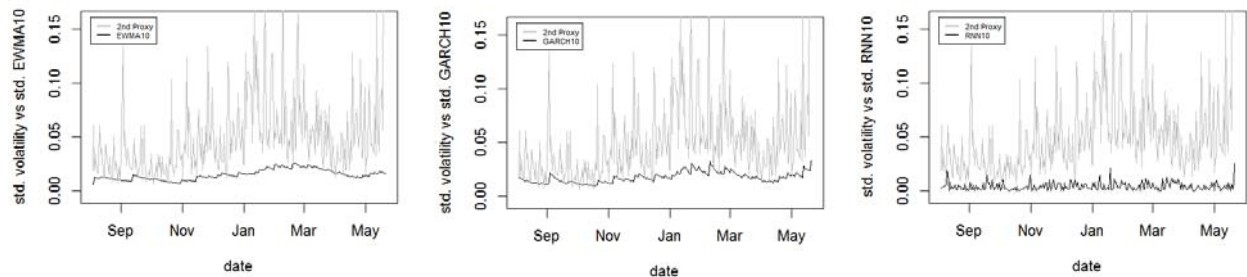

**Figure A4.** Out of sample standard deviation of realized volatility (2nd proxy) vs. standard deviation of 10 days ahead volatility forecasting of EWMA, GARCH and RNN model.

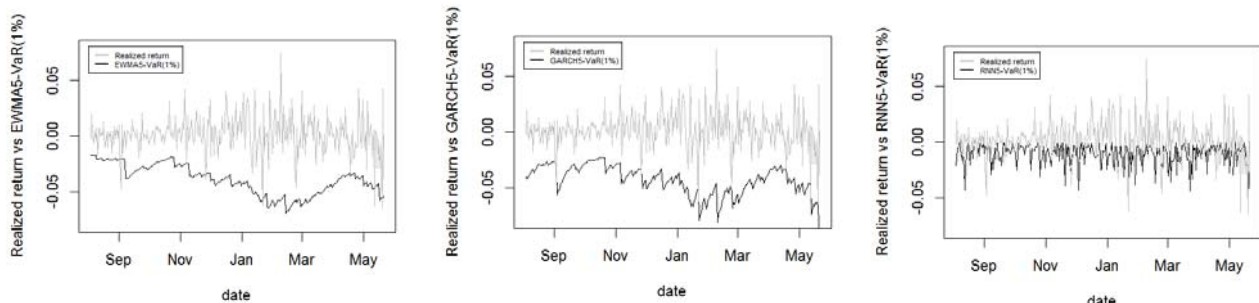

**Figure A5.** Out of sample realized return vs. 5 days ahead VaR (1%) of EWMA, GARCH and RNN model.

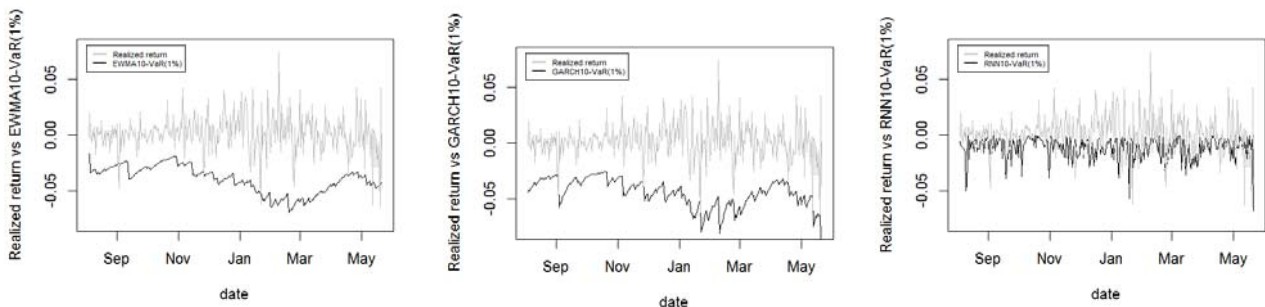

**Figure A6.** Out of sample realized return vs. 10 days ahead VaR (1%) of EWMA, GARCH and RNN model.



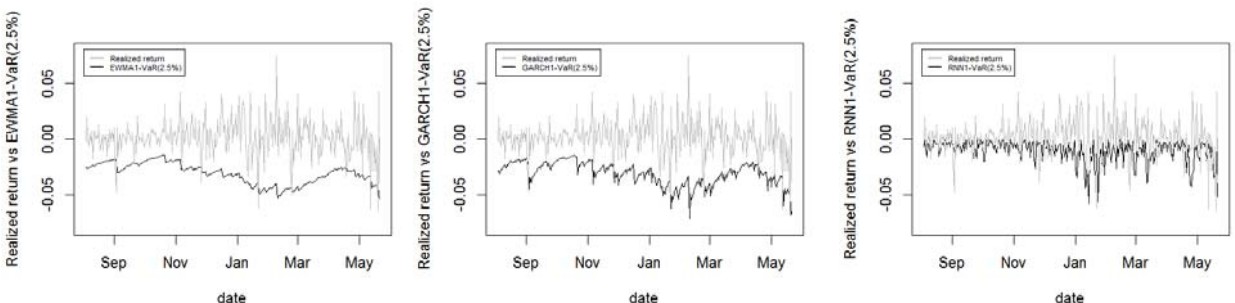

**Figure A7.** Out of sample realized return vs. 1 day ahead VaR (2.5%) of EWMA, GARCH and RNN model.

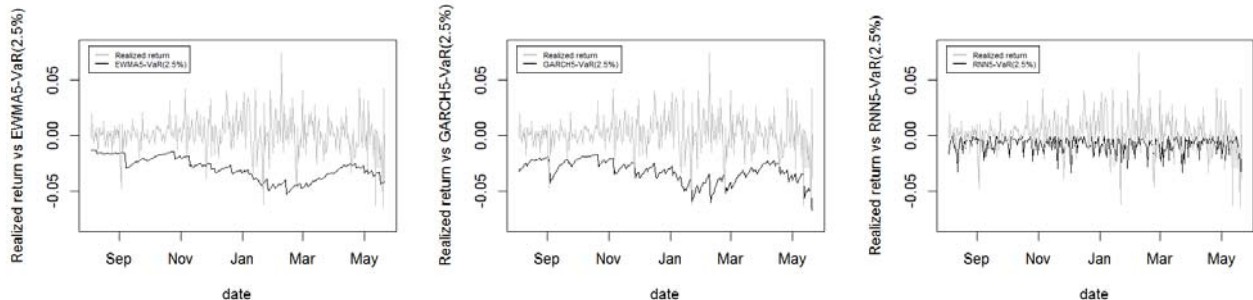

**Figure A8.** Out of sample realized return vs. 5 days ahead VaR (2.5%) of EWMA, GARCH and RNN model.

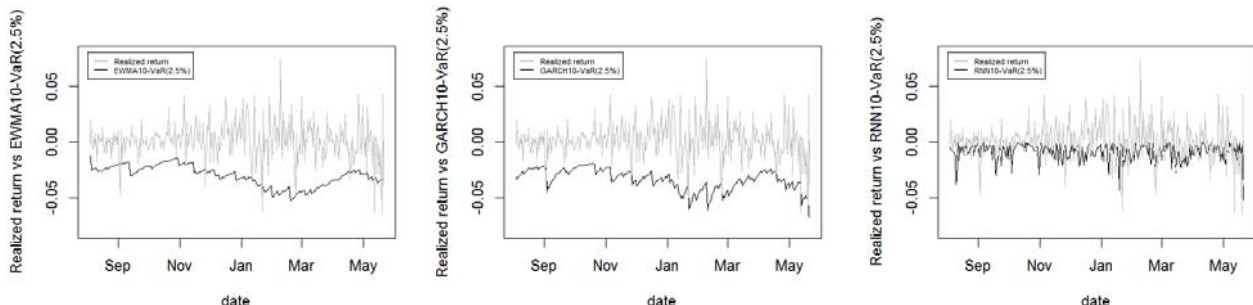

**Figure A9.** Out of sample realized return vs. 10 days ahead VaR (2.5%) of EWMA, GARCH and RNN model.

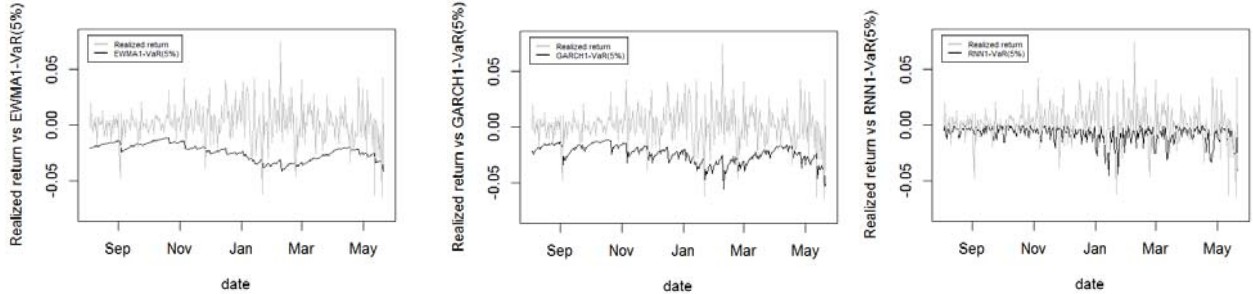

**Figure A10.** Out of sample realized return vs. 1 day ahead VaR (5%) of EWMA, GARCH and RNN model.

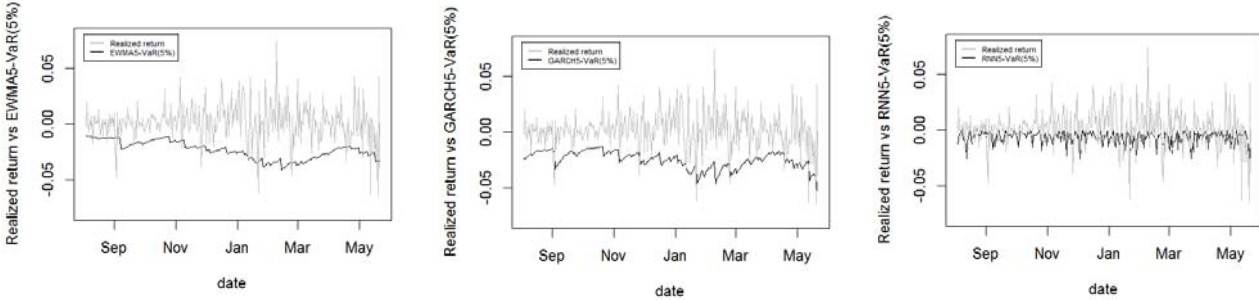

**Figure A11.** Out of sample realized return vs. 5 days ahead VaR (5%) of EWMA, GARCH and RNN model.

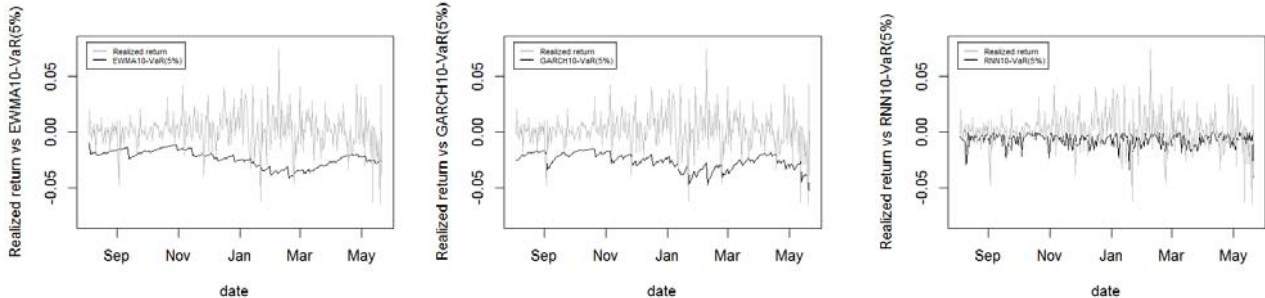

**Figure A12.** Out of sample realized return vs. 10 days ahead VaR (5%) of EWMA, GARCH and RNN model.

## Notes

[1] The standard deviation of the first and second proxy and the standard deviation of the 5 days ahead and 10 days ahead volatility forecasting of each model are presented in Appendix A.

[2] The figures of realized return and 5/10 days ahead VaR (1%) of each model are presented in Appendix A.

[3] The figures of realized return and 1/5/10 day(s) ahead VaR (2.5% and 5%) of each model are presented in Appendix A.

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
