# Peer review of "Bitcoin Return Volatility Forecasting: A Comparative Study between GARCH and RNN"

_jrfm, doi:10.3390/jrfm14070337_

Round 1

Reviewer 1 Report

In the article, the authors discuss the topic of bitcoin price volatility modeling. They compare the results of econometric models with neural networks. Unfortunately the authors only consider daily data, therefore the dataset is quite short. Additionally, it ends in November 2018.Why the authors did not extend the analysis to the present time. After all, they use readily available daily data?

During the period that the authors are covering, there have been significant changes in the cryptocurrency market. The volume and frequency of transactions increased significantly, which affected the fluctuation characteristics such as return distributions, volatility clustering and even multifractal effects - see Cracow group work: Physics Reports 2021 https://doi.org/10.1016/j.physrep.2020.10.005. Thus the results from 2013-2016 and 2017-2018 are hardly comparable. This may affect the parameters of the model as well as the learning process of the neural network.

The authors also test their models only on one, arbitrarily chosen, period of time - 2nd half of 2018 which was characterized by a fairly low volatility. 
In conclusion, I recommend to extend the range of data set up to 2021 and additional tests of the proposed models on a different data range, not just on one test set.

Author Response

We appreciate your valuable comments, and they are helpful for us. We have expanded the data set to May 2021. The ratio of in-sample data and out-of-sample data is about 9 to 1.

Reviewer 2 Report

The paper is very interesting, as it highlights one important issue: a type of neural network (RNN) fails in predicting VaR, while classical GARCH models are doing a better job.

Two questions:

 1. Maybe other NN models, like LSTM, CNN, or maybe a neural GARCH approach are better than RNN?

2. In backtesting VaR, we suggest to use some classical tests like Kupieck and Cristopherssen.

Author Response

We appreciate your valuable comments, and they are helpful for us.

1. The functionality of LSTM and GRU are similar, but GRU is computationally more efficient than LSTM; CNN works well for spatial information, but we are exploring temporal information, which RNN is more suitable for. Incorporating a GARCH model into the neural network is a great idea. However, in this study, we intend to compare the performance of the conventional economic model with the temporal neural network (RNN) to show a clear preference. However, it worth discussing whether combining them would perform better.

2. We would like to follow your suggestion of using the Kupieck and Cristopherssen test to evaluate the VaR performance. We will use them in the revised manuscript.

Reviewer 3 Report

First of all, congratolations to the authors for the paper wich is of great
scientific interest. Neural Networks applied to cryptocurrencies is
a great step to merge the economy with artificial intelligence.
From my point of view, I have nothing to object to this paper.
It has an excellent presentation, a good bibliographic review,
within the little knowledge in this ambit and robust results. Finally,
a clear discussion that leaves open a very interesting line of research.
Congratulations again.

Author Response

We appreciate your valuable comments, and they are helpful for us.

Reviewer 4 Report

  1. The title is faulty. You are doing a “horse race” between an econometric tool and a machine learning tool. The title should be something like this: “Bitcoin  Volatility Forecasting: A Comparative Study between GARCH and RNN”.
  2. Why GARCH is an economic model? It is an econometric model; it becomes an economic model if the variables of interest are economic and the model is structural. GARCH does not use any economic background besides being developed by Econometricians. In fact, what you are doing is comparing a parametric model with a non-parametric model. By the way, GARCH is a non-linear model.
  3. The literature review is amazingly shallow. The topic has received much attention in the literature. If you put the following keywords “volatility forecasting” and bitcoin in the academic google you obtain more than 500 entries. About Machine Learning (ML) applied to cryptocurrencies please read:
  • Fang F, Ventrea C, Basios M, Kong H, Kanthan L, Martinez-Rego D, Wub F, Li L (2020) Cryptocurrency trading: A comprehensive survey. arXiv preprint arXiv:2003.11352.
  • Sebastião, H. & Godinho, P. (2021). Forecasting and Trading Cryptocurrencies with Machine Learning under Changing Market Conditions. Financial Innovation, 7(1): 3.
  • Ji S, Kim J, Im H (2019) A Comparative Study of Bitcoin Price Prediction Using Deep Learning. Mathematics, 7(10), 898.
  1. Are you also using a moving window for GARCH? Are you revising the parameters of the GARCH model in a moving window? If not your methodology is biased in favor of the nonparametric model.
  2. Why an RNN? There are other methods. What is the motivation to use this tool?
  3. Why the Garman-Klass range volatility estimator? The open and close prices are artificial as the market is 24/7?
  4. Bitcoin was presented in 2008. Bitcoin has transaction costs, sometimes quite high. First names of authors? Susan Athey (2018)? Sean, Jason, and Simon (2018)? (where is this paper?). “Machine Learning methods are more advanced than traditional economic models in time series forecasting theoretically and empirically.” Theoretically, while these methods are data-driven? ML methods preserve more temporal information of a time series during training. Than what? Econometric models? An SMA? Why not a WMA which the most common to forecast volatility? The most used proxy for volatility is the bitcoin daily squared return.? Are you sure? Sorry to disagree.

Author Response

We appreciate your valuable comments, and they are helpful.

1&2. We would like to follow your suggestion and change the title to “Bitcoin Return Volatility Forecasting: A Comparative Study between GARCH and RNN”

3. We will improve the literature work to make it more sufficient.

4. We couldn’t use a rolling window on RNN. Because the RNN strategy is to train on in-sample data and test on out-of-sample data. Therefore, we have to give up a rolling window in GARCH to ensure apples to apples.

5. As we know, RNN is well known for its capability in capturing temporal information. The first paragraph in section 2.2 has mentioned that RNN has memory units, which favors in dealing with sequence problems, which is more suitable for time series prediction than other machine learning methods.

6. Garman and Klass volatility estimator is range-based volatility using four data points. It is an extension to Parkinson’s volatility and is easy to apply. It is a good point on the open/close price. The open price refers to the earliest data in range (UTC), and the closing price is the latest data in range (UTC).

7. The paperA Peer-to-Peer Electronic Cash System” was proposed in 2008, and the bitcoin network was launched in 2009.

Athey, S. "The impact of machine learning on economics." The Economics of Artificial Intelligence: An Agenda. Anonymous University of Chicago Press, 2018.

McNally, S., J. Roche, and S. Caton Predicting the price of bitcoin using machine learningIEEE, 2018.

The machine learning model is proposed in a more general scope that considers both linear and nonlinear features. It also preserves more temporal information of a time series during training.” Then “Machine Learning methods are more advanced than traditional economic models in time series forecasting theoretically...”

WMA is the most common one, but SMA is the most straightforward one used as a benchmark here.

We mean the most common proxy for financial asset volatility is the squared return (close to close) when only daily closing price is available, see Cumby et al. (1993) and Figlewski (1997). It is usually used as a benchmark.

Reviewer 5 Report

Bitcoin Return Volatility Forecasting: A Comparative Study of GARCH Model and Machine Learning Model

This paper assesses expected volatility of bitcoin using simple average, GARCH and machine learning models and points to the superiority of the last type of models which are data rather than economic based. Also, the paper assesses all these models in terms of estimating value at risk for risk management purposes.

Although the topic is relevant, the existing literature on the topic is quite rich. The authors however provide only a very brief description of the literature. This makes it also difficult to highlight the novelty of the current study. This is an area which should be improved in my opinion. Also, the introductory part could do more to explain the importance of the topic studied.

The empirical analysis related to the volatility estimation through GARCH and machine learning models seems to be well executed and also the assessment of model performance looks fine.  

However, it is very surprising to me that, despite being better in terms of volatility prediction, the machine learning models perform that bad in terms of value at risk. I wonder if there is not an error somewhere. The authors should provide more details about the VaR computation.

The conclusion part could focus more on the policy implications of the research.

The paper is coherent and easy to read. However, clearly placing the research in the literature emphasizing the novelty and revisiting the aspects related to VaR are elements which need to be dealt more in detail in my view.

Author Response

We appreciate your comments, and they are helpful for us. We will improve the literature work to make it more sufficient. Also, we will explain the importance of this topic more clearly.

It is a good idea to involve more policy implications of the research.

We are pretty confident that there is no error in VaR computation. And we will provide more details about it in the revised manuscript.

Round 2

Reviewer 1 Report

Extending the data set to the present time was only one point in my review.
What with the other issues?
1. Changes in the crypotocurrency market during considered time - see see Cracow group work: Physics Reports 2021 https://doi.org/10.1016/j.physrep.2020.10.005
2. Performing a test of the model in several selected periods with different regimes - bull/bear market and comparing the testuls. Currently, the model has only been tested in a single period 03.08.2020 - 21.05.2021, which is strong bull market and thus results may be biased.

I don't see references list in current manuscript version.

Author Response

Again, we appreciate your valuable comments. This study intends to give generalized guidance to investors of which approach is more suitable in practice, considering the market fluctuations conditions, but regardless of bull or bear market. Nevertheless, it is a great suggestion to test different periods according to the bull and the bear markets in future studies. The reference list is in the back matter file.

Reviewer 4 Report

Most of the issues raised in my report were addressed by the authors. Not all though. Nevertheless, I am comfortable with suggesting publication in the present form.

Author Response

We appreciate your valuable comments.

Reviewer 5 Report

The current version of the research is a step forward. The authors made additions and improvements in the methodological part, added some extra references and provided some comments regarded my objection related to VaR. I still consider that the empirical part is reasonably well executed.

However, the authors have not addressed completely my comments related to the literature review and the placing of their research within the literature. Although the literature on the topic is quite rich, it is still quite poorly reflected in the current version of the paper. Because of this issue the novelty of the current study is not clearly highlighted. Improvements in these areas would enhance the value of the current research.

Author Response

Again, we appreciate your valuable comments. We follow your suggestions by enriching the literature review, placing our research within the literature, and highlighting the novelty of the current study.

Round 3

Reviewer 1 Report

Now in the pdf file there is no title, abstract and reference list. I see the reference list in author_response.docx

Please make the correct citation: 

WÄ…torek, M.,  Drożdż, S., KwapieÅ„, J.,  Minati, L., OÅ›wiÄ™cimka, P. and  Stanuszek, M. "Multiscale characteristics of the emerging global cryptocurrency market." Physics Reports 901 (2021), doi 10.1016/j.physrep.2020.10.005

Author Response

We made the correct citation as you suggested.